# INVISIBILITY STICKERS AGAINST LiDAR: ADVERSARIAL ATTACKS ON POINT CLOUD INTENSITIES FOR LiDAR-BASED DETECTION

## ABSTRACT

Point cloud detection is crucial in applications such as autonomous driving systems and robotics. These systems utilize onboard LiDAR sensors to capture input point clouds, consisting of numerous three-dimensional coordinates and their corresponding intensity of laser reflection. Recent studies have proposed various adversarial schemes to highlight the vulnerability of point cloud detectors. However, these studies primarily focused on generating or perturbing the coordinate positions of input points and are hard to attack in the physical world, while largely overlooking the significance of their intensity. Through our exploration, we found that perturbing point cloud intensity poses significant security risks for point cloud object detectors. To the best of our knowledge, we are the first to attack on point cloud intensity and we propose an effective adversarial attack scheme, named I-ADV. Our method employs a voxel partition scheme to enhance physical implementation. To boost attack performance, we incorporate a gradient enhancement technique using 3D angle and distance features, along with an extremum-based gradient fusion strategy. Extensive experimental results demonstrate that by altering only point cloud intensity, our approach achieves state-of-the-art performance across detectors with various input representations, attaining attack success rates between 83.9% and 99.1%. Comprehensive ablation studies confirm the effectiveness and generality of the method's components. Additionally, comparing different attack schemes underscores the advantages of our point cloud intensity attack method in both performance and real-world applicability.

## 1 INTRODUCTION

The LiDAR-based point cloud object detectors are extensively used in applications such as autonomous driving systems and robotics (Guo et al. (2021)). They rely on onboard LiDAR sensors to capture precise point clouds as inputs, which typically consist of three-dimensional coordinates and their corresponding point cloud intensity of the laser reflections. The point cloud intensity indicates the strength of reflectivity of the objects' surfaces (Sun et al. (2020)), and it has become essential information for various tasks like road detection (Caltagirone et al. (2017)), segmentation (Tatoglu & Pochiraju (2012)), localization (Aldibaja et al. (2017)), and so on.

Recent studies (Szegedy et al. (2014); Goodfellow et al. (2015)) have shown that deep learning models are vulnerable to adversarial attacks. Attackers can induce errors in the system by adding customized perturbations to the input data, posing significant security risks in scenarios such as autonomous driving. Research on the adversarial vulnerability of point cloud detection can be broadly categorized into two types. The first type focuses on sensor-level attacks (Cao et al. (2019a; 2023); Jin et al. (2023)), where attackers exploit the working principles of LiDAR sensors by forging laser echo signals to create or hiding point clouds. The second type involves algorithm-level attacks, where attackers optimize corresponding adversarial samples targeting the detection algorithms. They obtain the coordinate perturbations by designing an IOU (Intersection over Union) loss (Cai et al. (2020); Wang et al. (2021)), maximizing LiDAR detection runtime (Liu et al. (2023)) and searching for the critical adversarial locations (Zhu et al. (2021)). Additionally, some studies utilize a differentiable LiDAR rendering algorithm (Möller & Trumbore (2005)) to optimize a universal adversarial object, aiming to hide target cars (Tu et al. (2020)), obscure obstacles (Cao et al. (2019b)), or create fake targets (Yang et al. (2021)).

However, to the best of our knowledge, existing research on adversarial vulnerabilities has primarily focused on manipulating the position coordinates of point cloud data, overlooking the widely used reflectivity intensity. First, our exploration indicates that replacing the original point cloud intensity of vehicles with random noise leads to an average of 34.5% of objects becoming undetectable across different detectors (see Table 1). This reveals that point cloud intensity is crucial for detection performance, and even random interference can significantly hinder recognition. Furthermore, unlike traditional adversarial attacks that disrupt the geometric shape of objects by modifying point cloud coordinates, developing an efficient attack method that relies solely on altering point cloud intensity presents a novel challenge. Lastly, point cloud intensity adversarial attacks possess a distinct advantage in terms of physical realizability, as attackers can

Figure 1: Illustration of attacks on point cloud intensity. (a) LiDAR sensors emit laser beams, which are reflected by objects, capturing both the 3D coordinates and intensity data. (b) In coordinate attacks, the 3D positions of the point clouds are altered while their intensities remain unchanged, whereas intensity attacks do the opposite. The color gradient from cool to warm represents point cloud intensity, ranging from low to high. (c) A real world example of perturbing point cloud intensity, showing that electrical tape reduces intensity, while reflective tape increases it. The point cloud data were captured by RS-Helios 1610 LiDAR[1].

change the point cloud intensity by applying different materials to an object's surface. In contrast, perturbing point cloud coordinates is much harder to achieve physically. We illustrate the attacks on intensity in Figure 1. It's worth noting that LiDAR sensors can easily produce a wide range of intensity values when scanning surfaces of common objects. For instance, as shown in Figure 1(b) and 1(c), the license plate and reflective tape used in traffic cones can register as maximum intensity, while the other parts of the vehicles register as lower intensity and it is common to reach the minimum intensity. This further underscores the physical feasibility of point cloud intensity attacks. Therefore, point cloud intensity adversarial attacks are both meaningful and challenging.

In this paper, we introduce the first adversarial attack scheme targeting point cloud intensity, namely I-ADV. To enhance physical realizability, we design a new optimization scheme based on voxel partitioning. To boost performance, we incorporate 3D angle features and distance features to enhance the gradient of each point cloud. Additionally, we design an extremum-based gradient fusion strategy to update the point cloud intensity within each voxel. Extensive experimental comparisons show that I-ADV achieves state-of-the-art attack performance against various baseline algorithms across different detector input representations, with an impressive attack success rate of up to 99.1%. This highlights the significant security threats posed by point cloud intensity attacks in real-world scenarios. In summary, we make the following three contributions:

- Unlike previous works that modify point cloud coordinates, we are the first to demonstrate that altering point cloud intensity can significantly impact the detection capabilities of point cloud object detectors, highlighting the importance of addressing adversarial robustness concerning point cloud intensity.

- We introduce I-ADV, the first attack scheme targeting point cloud intensity. This approach ensures physical realizability through a voxel partitioning-based scheme, along with novel gradient enhancement techniques utilizing 3D angle and distance features, and an extremum-based gradient fusion strategy to boost the performance.

- Extensive experiments validate our algorithm's superior attack performance across various input representatives of detectors. Comprehensive ablation studies confirm the effectiveness of our method from multiple perspectives, e.g., component contributions and transferability. Additionally, discussions of different types of attack algorithms underscore the advantages of point cloud intensity attacks in terms of performance and real-world applicability.

The remainder of this paper is organized as follows. In Section 2, we provide a brief introduction to the background knowledge on adversarial examples and vulnerability study of LiDAR-based detectors. Section 3 offers a detailed description of the proposed method. Section 4 represents the experiments conducted and analyzes the corresponding results. Section 5 provides more discussions of our work. Finally, we present the conclusions in Section 6.

## 2 RELATED WORK

### 2.1 ATTACKS ON IMAGES

Adversarial attacks on image-based deep learning models exploit vulnerabilities through small, often imperceptible perturbations. One of the earliest attacks is the Fast Gradient Sign Method (FGSM) (Goodfellow et al. (2015)), which perturbs images along the gradient of the loss function concerning the input image. Iterative Fast Gradient Method (I-FGSM) (Kurakin et al. (2018)) further improves FGSM by using multiple iterations, while MI-FGSM (Dong et al. (2018)) adds momentum to stabilize updates and escape local maxima. The Projected Gradient Descent (PGD) (Madry et al. (2018)) enhances the iterative attacks by introducing random initialization of perturbation. Adversarial attacks

---

[1]Refer to `https://www.robosense.ai/en/rslidar/RS-Helios`

are categorized into white-box attacks (Li et al. (2021a)) and black-box attacks (Chen et al. (2017)), based on the level of access the attacker has to the target model. Of these two classifications, white-box attacks enable the generation of highly effective adversarial examples by directly manipulating the model's gradients to maximize prediction errors, while black-box attacks rely on querying the model or exploiting transferability from surrogate models. The adversarial techniques have been applied across tasks like image classification (Moosavi-Dezfooli et al. (2016)), object detection(Song et al. (2018)), and segmentation(Xie et al. (2017)).

### 2.2 ATTACKS ON POINT CLOUD DETECTORS

**Sensor-level attacks.** Cao et al. (2019a) formulates the control strategy of spoofed points as an optimization problem to deceive the machine learning model through sensor attacks. In another work, Cao et al. (2023) makes the obstacles undetectable by leveraging the inherent automatic transformation and filtering processes of LiDAR sensors. Jin et al. (2023) explores the possibility of physically deceiving LiDAR-based detectors by injecting recorded or optimized point clouds using lasers.

**Algorithm-level attacks.** Some studies achieve the attack by focusing on the modification of point cloud coordinates. For instance, Wang et al. (2021) achieves coordinate perturbation for the car category by using an IOU (Intersection over Union) loss and imperceptible loss, significantly degrading the performance of the detection models. Liu et al. (2023) addresses the non-differentiable issue for the detectors and designs a novel loss to minimize the modifications while maximizing the runtime of the detection pipeline. Wang et al. (2023) spoofs fake obstacles at arbitrary locations by perturbing point cloud coordinates along each direction of the laser ray. Zhu et al. (2021) generates critical adversarial locations to fool the LiDAR perception system by designing heuristic location probing and location selection algorithms. Moreover, other studies insert a mesh object by a differentiable LiDAR rendering algorithm (Möller & Trumbore (2005)) and optimize the object's shape. For example, Cao et al. (2019b) explores the vulnerabilities of an industrial-level LiDAR-based autonomous driving system by proposing an optimization-based approach to generate 3D printable adversarial objects. Tu et al. (2020) generates a universal adversarial object that is equipped on the rooftop of target vehicles and makes them undetectable. Yang et al. (2021) optimizes a roadside adversarial object that will be recognized as vehicles invading the lane. However, there is currently no dedicated attack scheme targeting point cloud intensity. Due to the distinct nature of modifying intensity as the attack target, developing an effective new method is urgently needed. This paper addresses that gap.

## 3 METHODOLOGY

### 3.1 THREAT MODEL AND FORMULATION

Point cloud detectors create accurate 3D maps of the surrounding environment by leveraging the point cloud data from LiDAR sensors. A point cloud sample $\boldsymbol{X}$, typically consists of spatial and intensity information of $N$ points:

$$\boldsymbol{X} = \{P_n | P_n = (p_{coor}, p_i)_n, n = 1, ..., N\}, \tag{1}$$

where $P_n$ is an element point, $p_{coor}$ is its three-dimensional coordinates, and $p_i$ is its reflection intensity. Then, the adversarial point cloud $\boldsymbol{X^*}$ based on intensity perturbation $\boldsymbol{\delta}$ can be formulated as:

$$\begin{aligned} \boldsymbol{X^*} &= \boldsymbol{X} + \boldsymbol{\delta} \\ &= \{P_n | P_n = (p_{coor}, p_i + \delta_i)_n, n = 1, ..., N\}. \end{aligned} \tag{2}$$

That is, the modifications focus on the intensity, while the spatial coordinates of the point cloud remain unchanged. We formulate the purpose of the intensity attacks as the following optimization problem:

$$\mathcal{F}(\boldsymbol{X}) \neq \mathcal{F}(\boldsymbol{X^*}), \ s.t. \ \mathcal{D}(\boldsymbol{X} - \boldsymbol{X^*}) < \eta, \tag{3}$$

where $\mathcal{F}$ is the output of the detection model, $\mathcal{D}$ is the distance between the original and adversarial point clouds, and $\eta$ is the perturbation budget.

### 3.2 INTENSITY ADVERSARIAL ATTACK

In this paper, we present the first adversarial attack targeting point cloud intensity, named I-ADV, with the framework shown in Figure 2. We detailed the introduction of the key parts in the following.

**Voxel Partition.** To optimize intensity attacks, one might consider adjusting the intensity information of point clouds point by point, like FGSM and PGD. Apparently, point-by-point modifications are difficult to implement in the real world. However, as shown in Figure 1(c), attackers can change the reflection intensity by applying different materials to various regions of the object's surface based on our exploration. Therefore, in our scheme, the point cloud sample is first divided

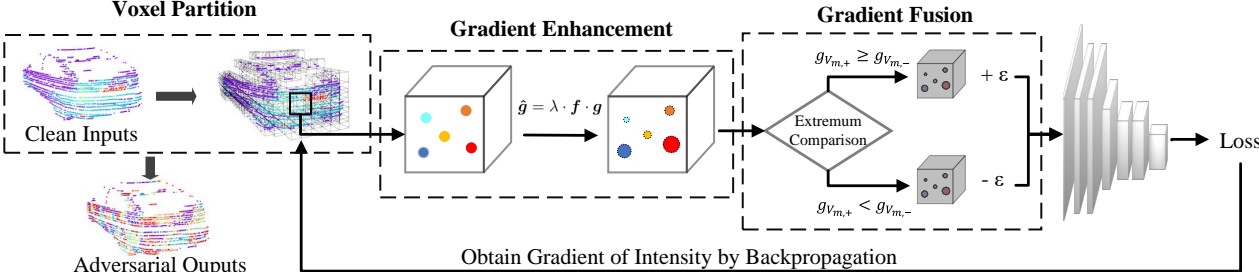

Figure 2: Overview of the proposed intensity attack scheme, I-ADV. Our method utilizes a voxel-based optimization framework, where all point clouds are initially allocated to their respective voxels based on voxel partition. The gradient of intensity within each voxel is then enhanced by reflection features $f$. Finally, all points within the voxel are updated in the same direction based on the comparison of extremums of the enhanced gradients.

into small equally sized cubes, also known as voxels, and all the points belong to the same voxel are updated in the same direction to ensure physical applicability. The voxel partition process can be formulated as:

$$\boldsymbol{V} = \{\boldsymbol{V_m}|m = 1, ..., M\} \xleftarrow{\text{Voxel Partition}} \{\boldsymbol{X}, V_{size}\}, \tag{4}$$

where $\boldsymbol{V}$ is a set of voxels, and $V_{size}$ and $M$ is the size and the number of voxel $\boldsymbol{V_j}$, respectively.

**Gradient Enhancement.** Based on our experiments (refer to Figure 1(b) and Figure 4), we have identified two patterns for the point cloud intensity: first, when the laser beams emitted by the LiDAR sensor strike the object's surface perpendicularly, the point cloud intensity is relatively high, whereas the intensity decreases significantly when the beams are nearly parallel to the surface. Second, as the distance between the object and the LiDAR sensor increases, the point cloud intensity tends to be weak. Building on these insights, we designed a reflectivity feature to boost the effectiveness of the attack on each point. The goal is to modify the point cloud intensity in a manner that intentionally disrupts these identified patterns.

To achieve this, firstly, we utilize the k-d tree algorithm to search for neighboring points for each point based on their 3D coordinates. Next, we perform plane fitting and estimate the normal vector of the surface using PCA and eigenvalue decomposition. Finally, we obtain the 3D angle $\phi$ between this normal vector and the direction of the laser beams. Secondly, we compute the Euclidean distance $\boldsymbol{d}$ from each point to the LiDAR sensor. Overall, the reflectivity feature $\boldsymbol{f}$ can be formulated by:

$$\boldsymbol{f} = \sin(\boldsymbol{\phi}) \cdot \sin(\frac{\boldsymbol{d}}{d_{MAX}} \cdot \frac{\pi}{2}) \tag{5}$$

where $d_{MAX}$ is the distance of maximum detection range, and $\sin(\cdot)$ is used as a normalized function. In this formula, the first term is the 3D angle feature that aims to disrupt the angular pattern of the point cloud intensity distribution, while the second term is the distance feature that targets the distance pattern. We obtain the enhanced gradient $\hat{\boldsymbol{g}}$ by weighting the original gradient $\boldsymbol{g}$ using the reflectivity features and a hyper-parameter $\lambda$ for adjustment:

$$\hat{\boldsymbol{g}} = \lambda \cdot \boldsymbol{f} \cdot \boldsymbol{g}, \tag{6}$$

**Gradient Fusion.** To determine the update directions $\{+, -\}$ for the points within a voxel, how to utilize their gradients is a crucial challenge. However, naive average or random strategies fail to fully exploit the magnitude information of the gradients (refer to Table 1). To address this, we propose a gradient fusion strategy based on extremum. We obtain the fusion result by comparing the absolute values of the maximum extremum and the minimum extremum for the gradients in a voxel:

$$g_{V_{m,+}} = |\max(\hat{\boldsymbol{g}}_{\boldsymbol{V_m}})|, g_{V_{m,-}} = |\min(\hat{\boldsymbol{g}}_{\boldsymbol{V_m}})|, \tag{7}$$

$$g_{V_{m,f}} = \begin{cases} +, & \text{if } g_{V_{m,+}} \geq g_{V_{m,-}} \\ -, & \text{otherwise} \end{cases}, \tag{8}$$

where $\hat{\boldsymbol{g}}_{\boldsymbol{V_m}}$ is a set of the gradients for voxel $\boldsymbol{V_m}$ after gradient enhancement, $g_{V_{m,f}}$ is the fusion result that denotes its update direction. The adversarial results for a voxel are updated by:

$$\boldsymbol{V_m^*} = \boldsymbol{V_m^*} + \varepsilon \cdot sign(g_{V_{m,f}}) \tag{9}$$

where $\boldsymbol{V_m^*}$ is the adversarial result for voxel $\boldsymbol{V_m}$. In the end, the adversarial point cloud $\boldsymbol{X^*}$ is obtained by reconstructing the adversarial voxels:

$$\boldsymbol{X^*} \xleftarrow{\text{Reconstruction}} \{\boldsymbol{V_m^*}|m = 1, ..., M\}. \tag{10}$$

---

**Algorithm 1:** Algorithmic process of I-ADV

---

**Input:** Raw input point clouds sample $\boldsymbol{X}$ with $M$ voxels; network weight $\theta$; ground true label $\boldsymbol{y}_{gt}$; number of iterations $T$.

**Input:** The size of perturbation $\varepsilon$; iterations $T$; decay factor $\mu$; hyper-parameter $\lambda$.

**Output:** Adversarial point clouds $\boldsymbol{X}^*$.

1: Calculate the 3D angle $\phi$ and distance $\boldsymbol{d}$ for point cloud $\boldsymbol{X}$
2: $\boldsymbol{f} = \sin(\boldsymbol{\phi}) \cdot \sin(\frac{\boldsymbol{d}}{d_{MAX}} \cdot \frac{\pi}{2})$ // Formula (5)
3: $\boldsymbol{g}_0 = 0, \boldsymbol{X}_0^* = X$
4: **for** $t = 0$ to $T - 1$ **do**
5:     $\boldsymbol{V}_t = \{\boldsymbol{V}_{t,m} | m = 1, ..., M\} \xleftarrow{\text{Voxel Partition}} \{\boldsymbol{X}_t^*, V_{size}\}$ // Formula (4)
6:     $\boldsymbol{g}_{t+1} = \mu \cdot \boldsymbol{g}_t + \frac{\nabla_{\boldsymbol{X}} L(f(\theta, \boldsymbol{X}_t^*, \boldsymbol{y}_{gt})}{\|\nabla_{\boldsymbol{X}} L(f(\theta, \boldsymbol{X}_t^*, \boldsymbol{y}_{gt}))\|}$
7:     $\boldsymbol{g}_{t+1} = \lambda \cdot \boldsymbol{f} \cdot \boldsymbol{g}_{t+1}$ // Formula (6)
8:     **for** $m = 1$ to $M$ **do**
9:         Obtain result of gradient fusion $g_{V_{m,f}}$ // Formula (7) and (8)
10:        $\boldsymbol{V}_{t+1,m}^* = \boldsymbol{V}_{t,m}^* + \varepsilon \cdot sign(g_{V_{m,f}})$ // Formula (9)
11:     **end for**
12:     $\boldsymbol{X}_{t+1}^* \xleftarrow{\text{Reconstruction}} \{\boldsymbol{V}_{t+1,m}^* | m = 1, ..., M\}$ // Formula (10)
13: **end for**
14: **return** $\boldsymbol{X}_T^*$

---

To clarify the algorithmic process, we present the pseudo-code of the proposed scheme in Algorithm 1. As shown in Algorithm 1, the reflectivity features can be pre-processed to enhance computational efficiency since they are inherently linked to the 3D coordinates. We utilize the framework of MI-FGSM to access the gradients during iterative updates.

## 4 EXPERIMENTS

### 4.1 EXPERIMENTAL SETUP

**Datasets and Target Models.** The KITTI dataset (Geiger et al. (2012)) is a widely-used benchmark for autonomous driving research, offering 3712 training samples, 3769 validation samples, and 7518 test samples. It includes LiDAR point clouds along with 3D bounding box annotations for various objects at three difficulty levels: Easy, Moderate, and Hard. To ensure comprehensive coverage, we selected three types of both classic and recent point cloud object detectors as our target networks: (1) point-based detectors, including PointRCNN (Shi et al. (2019)) and IA-SSD (Zhang et al. (2022)); (2) voxel-based detectors, such as PointPillar (Lang et al. (2019)) and Voxel R-CNN (Deng et al. (2021)); and (3) point-voxel-based detectors, including PV-RCNN (Shi et al. (2020)) and PDV (Hu et al. (2022)).

**Evaluation Metrics.** Our evaluation metric is attack success rate (ASR), which measures the percentage of instances where a target object is initially detected but fails to be detected after the attack. In our experiments, we specifically target the "Car" class and treat each object within a scene as an individual sample. A car is considered successfully detected if the resulting 3D IoU (Intersection over Union) exceeds 0.7.

**Comparison Baselines.** We primarily consider two types of iterative attacks as comparison methods. The first category is point-wise modification attack, including uniform random noise, PGD (Madry et al. (2018)), MI-FGSM (Dong et al. (2018)), and NI-FGSM (Lin et al. (2020)). The second category focuses on voxel-wise modifications, where the voxel serves as the smallest modification unit, and the modification direction is determined through a gradient fusion strategy. These voxel-based approaches are further divided according to how the gradient fusion result is obtained, including (1) the Voxel-Random method, where the gradient of a randomly chosen point within the voxel is used as the fused gradient; (2) Voxel-Average method, which uses the average of all gradients within the voxel; and (3) Voxel-Voting method, where the update direction is decided by majority rule, based on counting the number of positive and negative values of all gradients within the voxel.

**Implementation Details.** In our experiments, to ensure a fair comparison, all attack methods were executed for 10 iterations. Note that the point cloud intensity ranges from $[0, 1]$. The intensity is updated by 0.2 per iteration, with a maximum perturbation limit of 1, as any intensity magnitudes within this range are commonly seen and achievable in the real world (refer to Figure 1). In our approach, the voxel size $V_{size}$ for the voxel partition is set to be a cube with an edge length of 0.1 meters, the decay factor $\mu$ is set to 1.0, the hyper-parameter $\lambda$ is set to 1000, and the maximum detection range $d_{MAX}$ is 75 meters.

Table 1: The attack success rates (%) on the KITTI *val* split for the "Car" category at the Moderate difficulty level. The best performance is marked with **bold**.

| Attack Algorithms | Victim Models | Point-based | | Voxel-based | | Point-voxel-based | |
|---|---|---|---|---|---|---|---|
| | | PointRCNN | IA-SSD | PointPillar | Voxel R-CNN | PV-RCNN | PDV |
| Point-wise Attacks | Random | 32.0 | 64.3 | 33.2 | 34.6 | 24.2 | 18.7 |
| | PGD (Madry et al. (2018)) | 41.6 | 83.9 | 73.2 | 69.8 | 62.6 | 76.1 |
| | MI-FGSM (Dong et al. (2018)) | 44.3 | 84.6 | 65.8 | 62.7 | 62.9 | 65.3 |
| | NI-FGSM (Lin et al. (2020)) | 47.7 | 83.0 | 42.4 | 69.0 | 73.4 | 69.1 |
| Voxel-wise Attacks | Voxel-Random | 45.1 | 83.8 | 62.7 | 60.4 | 61.5 | 62.8 |
| | Voxel-Average | 42.1 | 85.1 | 65.6 | 62.1 | 62.7 | 64.4 |
| | Voxel-Voting | 47.0 | 91.0 | 68.7 | 70.5 | 70.0 | 74.1 |
| | **I-ADV (Ours)** | **87.3** | **99.1** | **83.9** | **91.8** | **90.3** | **97.7** |

Table 2: Ablation study for different components of the proposed method. $ASR_E$, $ASR_M$ , and $ASR_H$ denote the attack success rate for PointRCNN at easy, moderate, and hard difficulty on KITTI *val* split, respectively. In cases where the extremum fusion strategy was not applied, we used the average fusion strategy for comparison.

| 3D angle feature | Distance feature | Extremum based fusion | $ASR_E$ | $ASR_M$ | $ASR_H$ |
|---|---|---|---|---|---|
| ✗ | ✗ | ✗ | $33.5_{0.0}$ | $42.1_{0.0}$ | $48.7_{0.0}$ |
| ✓ | ✗ | ✗ | $34.8_{+1.3}$ | $43.7_{+1.6}$ | $50.7_{+2.0}$ |
| ✗ | ✓ | ✗ | $35.7_{+1.2}$ | $44.0_{+1.9}$ | $50.8_{+1.3}$ |
| ✓ | ✓ | ✗ | $35.2_{+1.7}$ | $44.6_{+2.5}$ | $51.5_{+2.8}$ |
| ✗ | ✗ | ✓ | $80.6_{+47.1}$ | $86.8_{44.7}$ | $88.6_{+39.9}$ |
| ✓ | ✓ | ✓ | $81.1_{+48.1}$ | $87.3_{+45.2}$ | $88.9_{+40.2}$ |

## 4.2 EVALUATION RESULTS

The ASR (Attack Success Rate) evaluation results for different detection models under various attack algorithms are presented in Table 1. Based on these results, we make the following key observations: (1) Random noise method: Optimization-based methods significantly outperform the random noise approach, demonstrating that learning adversarial intensity distributions through algorithmic optimization provides a clear advantage. (2) Different detector architectures: The vulnerability of voxel-based and point-voxel-based detectors is relatively similar, but point-based detectors show polarized performance. Specifically, PointRCNN proves highly resilient to attacks, while IA-SSD is the most vulnerable, indicating that despite IA-SSD's efficiency in detection, it sacrifices robustness. (3) Point-wise vs. voxel-wise optimization: There is no clear winner between point-wise and voxel-wise methods. However, the voting-based method consistently outperforms other baselines, suggesting that intelligently fusing gradients enhances attack success. (4) Our method: Our approach stands out across all detection networks, achieving ASR between 87.3% and 99.1%, highlighting the effectiveness of our gradient enhancement and fusion strategy.

## 4.3 ABLATION STUDY

**Components of our method.** In our method, gradient enhancement techniques, including the utilization of 3D angle features and distance features, and the extremum-based gradient fusion strategy are the key components. Here, we separately validate these components as shown in Table 2. From Table 2, it is evident that using either 3D angle features or distance features alone results in an ASR improvement of 1.3–2.0% and 1.2–1.9%, respectively. When both features are combined, the improvement increases to 1.7–2.8%. Notably, the extremum-based gradient fusion strategy alone yields a substantial boost of 39.9–47.1%. When this strategy is combined with gradient enhancement, the ASR further increases to 40.2–48.1%. It indicates that the proposed fusion strategy is essential for achieving successful attacks, while gradient enhancement provides an additional performance boost.

**Voxel size and perturbation range.** In addition to gradient enhancement and fusion, we found that voxel size during partition and the maximum perturbation limit significantly impact attack performance. The ablation studies for these factors are shown in Figure 3. As demonstrated in Figure 3(a), smaller voxel sizes consistently lead to better ASR performance, suggesting that finer voxel granularity enhances optimization in our method. However, as voxel size increases (e.g., to 0.5 m), the ASR starts to plateau. Moreover, Figure 3(b) also shows that larger perturbation limits result in higher ASR performance, highlighting that perturbation amplitude is one of the key factors for adversarial attacks on point cloud intensity.

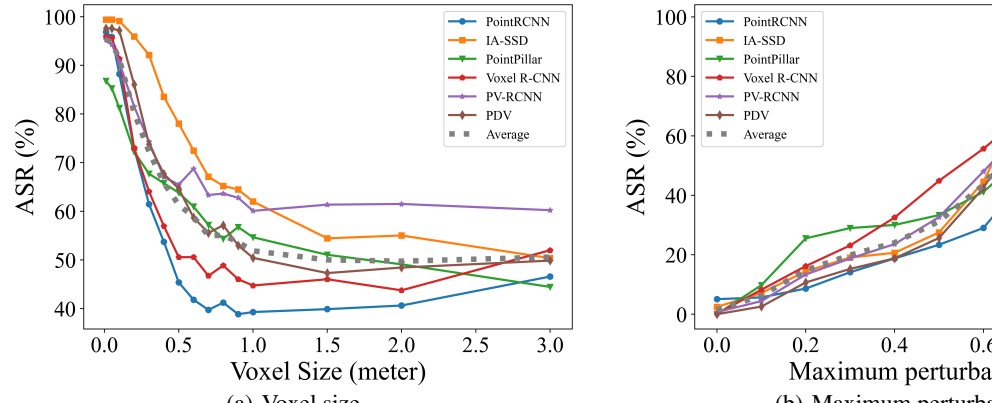

|  | (a) Voxel size | (b) Maximum perturbation |
|---|---|---|

Figure 3: Ablation studies for voxel size and maximum perturbation limit of point cloud intensity. To optimize computational efficiency, we randomly selected one-tenth of the samples from the KITTI *val* set and performed evaluations across various detectors. The gray dashed line in the figures represents the average performance across all detectors.

Table 3: Evaluation of transferability across different models. The attack success rates (%) on the KITTI *val* split for the "Car" category at Moderate difficulty are given. The best performance and performance of the white-box setting are marked with **bold** and an asterisk (*), respectively.

| Source \ Target | PointRCNN | IA-SSD | PointPillar | Voxel R-CNN | PV-RCNN | PDV |
|---|---|---|---|---|---|---|
| PointRCNN | 86.7* | 99.2 | 78.2 | 92.8 | 84.3 | 97.7 |
| IA-SSD | **87.9** | 99.3* | 79.4 | **93.2** | 85.7 | 97.8 |
| PointPillar | 85.5 | 98.5 | 83.8* | 89.6 | 84.8 | 97.4 |
| Voxel R-CNN | 86.1 | 98.8 | 81.0 | 91.8* | 86.1 | 97.8 |
| PV-RCNN | 87.5 | 98.8 | 82.7 | 92.1 | 90.3* | **97.9** |
| PDV | 87.4 | 99.1 | 81.7 | 92.8 | 87.6 | 97.9* |

**Transferability.** To assess the generalization performance of our method on new detectors, we conducted experiments using a transfer attack setup. We evaluated how adversarial samples generated by various source detectors performed against different target detectors, with the results presented in Table 3. Firstly, the table reveals that IA-SSD and PDV were particularly vulnerable to our method, with ASR reaching 98.9% and 97.7%, respectively, when used as target detectors. Moreover, the white-box attack setup delivered the highest success rates in most cases. Interestingly, adversarial samples generated by IA-SSD occasionally surpassed some white-box attacks, such as with PointRCNN and Voxel R-CNN, where ASR improved by 1.2% and 1.4%, respectively. We argue that this is because IA-SSD is better at exposing the weaknesses of samples, making its adversarial examples more potent. Overall, our method exhibited strong attack performance across various transferability scenarios, achieving ASR 78.2-99.2% in black-box settings.

**Visualization.** To qualitatively analyze the modification results and attack effects of different algorithms, we visualized the results for three samples in Figure 4. First, regarding the modification results, we observe that the point cloud intensity distribution of cars in the clean samples is related to both the surface angle and the distance of the point cloud (as detailed in Section 3). After applying the attack algorithms, this distribution is disrupted. For example, the point-wise algorithm PGD results in a chaotic appearance of point cloud intensity, while the voxel-wise algorithm yields a more segmented, block-like structure. In terms of the predictions before and after attacks, our method shows significantly improved effectiveness compared to baseline approaches, resulting in more missed detections. Interestingly, The attacks on point cloud intensity of cars cause not only misjudgments in vehicle direction but also misclassifications of background as cars (such as sample C during the I-ADV attack). This highlights that point cloud intensity attacks can be effective in a wider range of attack scenarios.

**Scene-level perturbation.** In the previous experiments, the perturbation of point cloud intensity attacks was confined to the target vehicles, i.e., object-level perturbations. In Figure 5, we further explore the performance under a full-scene interference setting, i.e., scene-level perturbations. The following key observations were made: First, regarding the attack algorithms, our method achieves the highest attack performance across all target detectors, with the ASR reaching up to 100%, strongly demonstrating the effectiveness of our approach. Besides, NI-FGSM shows the largest improvement, with an average increase of 24.0%, whereas PGD exhibits the smallest improvement, averaging only 3.8%. Interestingly,

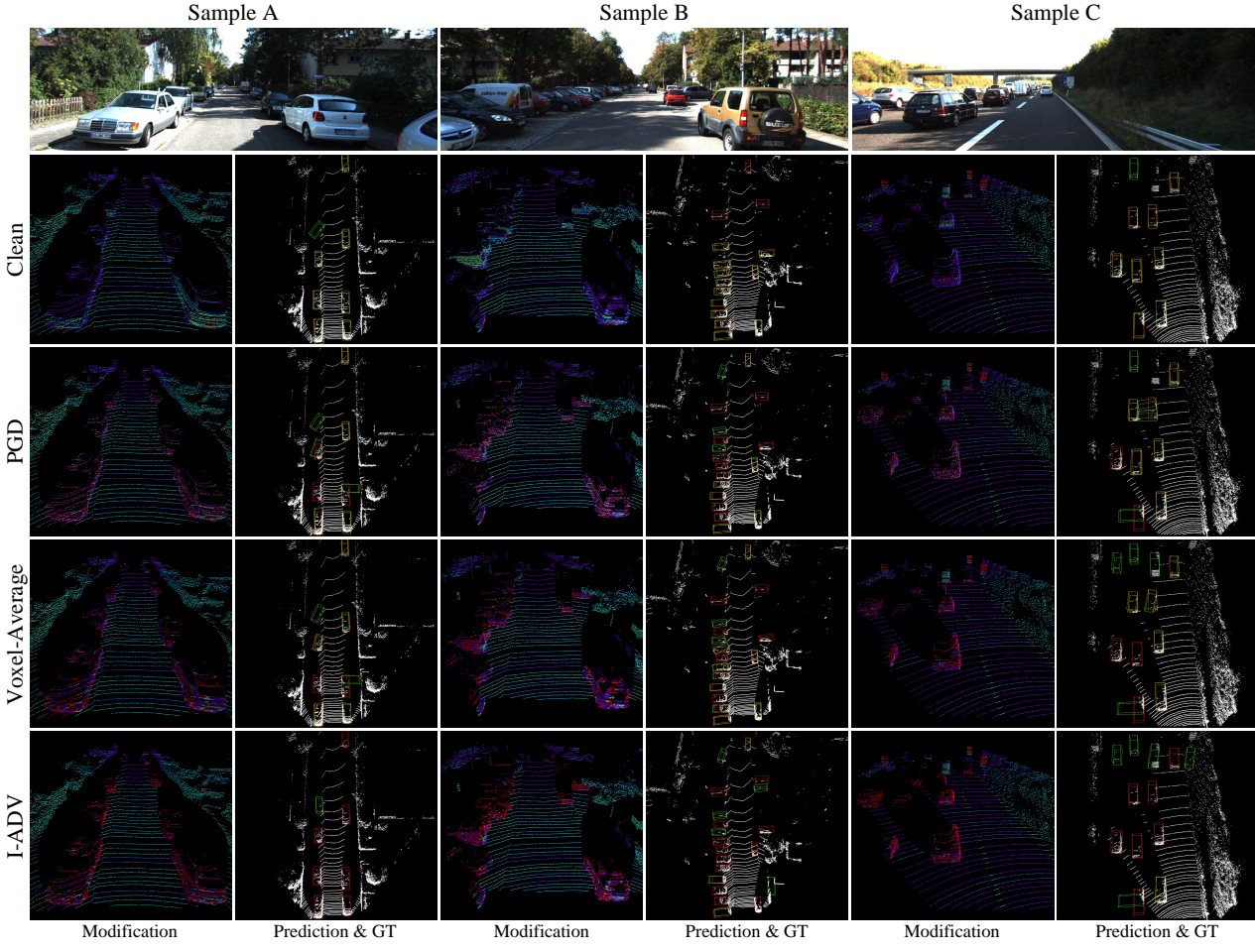

Figure 4: The visualization for the modifications results of the point cloud intensity and the predicted results before and after attacks on PointRCNN. For the point cloud intensity modification results, we map the intensity values from low to high onto a color gradient ranging from cool to warm colors (refer to Figure 1(b)). The attack results are shown with predicted detection boxes (in green) and ground truth boxes (in red).

PGD even experiences performance degradation on IA-SSD and PDV. Second, in terms of the detectors, the average ASR on IA-SSD reaches 92.3%, confirming it as the most vulnerable detector, consistent with the conclusions in Section 4.2. Overall, scene-level perturbations result in higher attack performance compared to object-level perturbations, with an average ASR improvement of 16.8%, revealing the greater attack potential of various point cloud intensity attack algorithms.

Table 4: Comparison of different types of attacks. Note that all results are obtained on the car category of the KITTI *val* split with PointRCNN, using the same evaluation metric (ASR).

| Algorithm | Attack Type | ASR (%) | Physical Realizability |
|---|---|---|---|
| Tu et al. (2020) | Insertion of a mesh object | 32.3 | 3D Printing |
| Cai et al. (2020) | Midification on point cloud coordinates | 79.4 | Infeasible |
| Wang et al. (2021) | | 82.8 | |
| **Ours** | Modification on point cloud intensity | **87.3** | Pasting |

## 5 DISCUSSION

**Comparison with various types of attacks.** As the first attack algorithm based on modifying point cloud intensity, we compared it with other types of attacks, as shown in Table 4. Note that all attacks were evaluated on the same dataset using the same evaluation metric (refer to section 4.1), thus we referenced the performance reported in their respective papers.

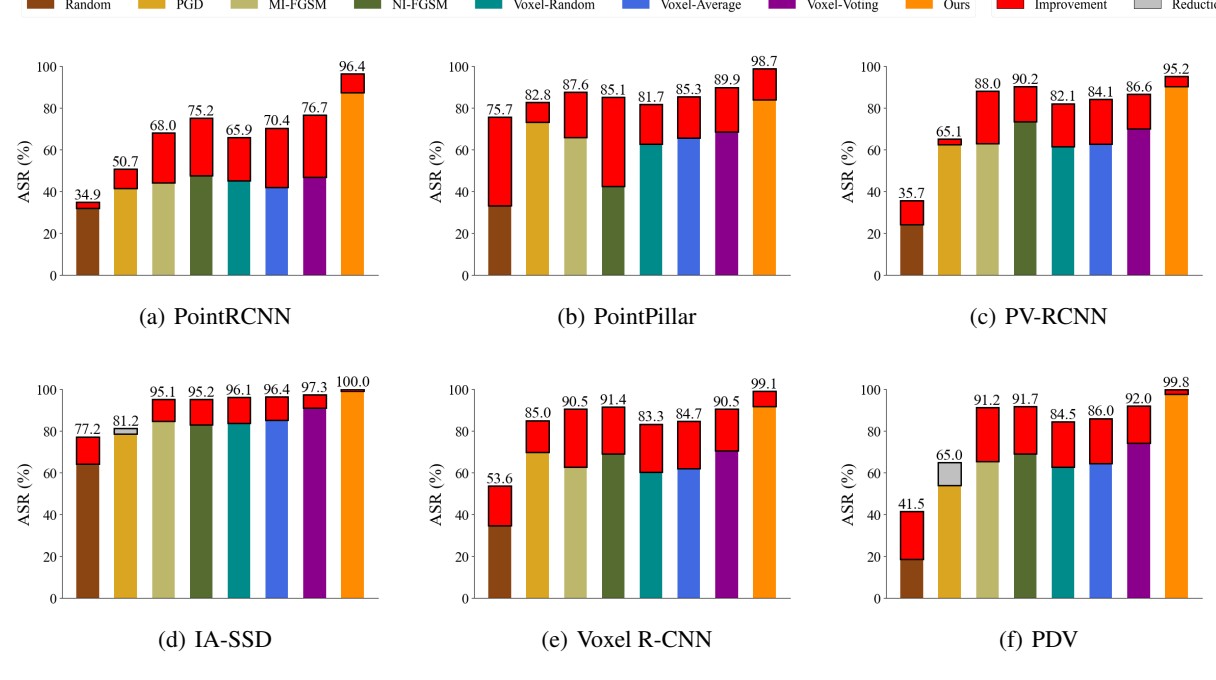

Figure 5: Performance evaluation for scene-level point cloud intensity adversarial attacks. The experimental outcomes were obtained using different attack algorithms and target detectors on the KITTI *val* split. The red and gray bars indicate the improvements and declines in performance compared to the results of object-level point cloud intensity adversarial attacks (Table 1), respectively.

From the table, two key observations can be made. First, in terms of performance, the method involving the insertion of a mesh object is the least effective, which we attribute to its partial augmentation of the target object's geometric profile, resulting in limited attack capability. On the other hand, attack methods based on modifying point cloud coordinates achieve better results, as they can fully disrupt the geometric shape of the target object, making it more difficult for the detector to recognize. Our method, by modifying point cloud intensity, demonstrates the strongest attack performance, which we argue is due to the detector's strong reliance on the distribution characteristics of point cloud intensity, making intensity modifications particularly impactful. Second, regarding physical applicability, the insertion of mesh objects requires the 3D printing of large adversarial objects, which is costly. Modifying point cloud coordinates, while effective, is difficult to achieve in a physical context. In contrast, our method can be realized by applying stickers with varying reflective intensity to the surface of objects, offering both physical feasibility and low cost.

**Limitations and future work.** Despite the innovation of our method, there are some unresolved issues as follows. (1) Physical exploration: We modeled the point cloud intensity attacks using physically feasible methods. While the intensity distribution generated by the attacks is broad, determining which materials can adjust the intensity by specific magnitudes in the real world requires further engineering exploration. (2) Limited attack variability: The performance of adversarial attacks under combined attacks targeting both intensity and coordinates, remains unexplored due to the lack of unified standards for perturbation magnitudes. (3) Focus on LiDAR-only systems: Multi-sensor fusion allows the system to leverage both the rich visual information from cameras and the precise spatial data from LiDAR. The potential for jointly optimizing adversarial perturbations across both image texture patches and point cloud intensity remains unexplored.

## 6 CONCLUSION

In this paper, we have highlighted the critical role of point cloud intensity in the adversarial attack on LiDAR-based object detectors, addressing a significant gap in the existing literature on adversarial vulnerabilities. By introducing I-ADV, the first attack scheme dedicated to modifying point cloud intensity, we have demonstrated the outstanding impact these alterations can have on detection capabilities. Our innovative optimization framework, which incorporates voxel partition, enhanced gradient techniques and gradient fusion strategy, achieves state-of-the-art attack performance. Extensive comparisons validate the effectiveness, generality, and real-world feasibility of our approach. This research emphasizes the urgent security threats posed by intensity-based adversarial attacks in real-world applications such as autonomous driving.

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
