# OpenReview forum: "Invisibility Stickers Against LiDAR: Adversarial Attacks on Point Cloud Intensity for LiDAR-based Detection"
_ICLR.cc/2025/Conference — ICLR 2025 Conference Withdrawn Submission_

### Official Review · Reviewer_dWMh · 2024-11-02

**Soundness:** 3
**Presentation:** 3
**Contribution:** 3
**Rating:** 6
**Confidence:** 5

**Summary:**

This work proposed an attack scheme targeting point cloud strength, combining voxel partitioning, gradient enhancement, and gradient fusion strategies to achieve advanced attack performance, proving the effectiveness and physical feasibility of point cloud strength attacks.

**Strengths:**

1. Experimental results show that the attack performance is relatively good.
2. Propose an attack method based on adversarial strength, which is different from most point based methods.
3. Good transferability, further discussion on scenario level interference.
4. Utilizing voxel segmentation and gradient enhancement fusion for attack.

**Weaknesses:**

1. The dataset is not introduced.
2.The paper proposes pasting stickers with different reflection intensities on the surface, but does not provide specific pictures and processes, only a small picture of a simple car sticker.
3. The paper mentions KITTI validation sets of different difficulty levels multiple times, but there are no examples to illustrate them.
4. The material of the sticker is not detailed, and this is important for readers, readers need to be clear about the mechanism of Lidar point cloud absorption or material source, which is conducive to the promotion of the work in the community.

**Questions:**

1. The authors should introduce the dataset in detail.
2. The paper proposes to paste stickers with different reflection intensities on the surface, and specific pictures and processes should be provided
3. More latest attack methods can be used.
4. The paper should provide an explanation for KITTI validation sets of different difficulty levels.
5. More clearly introduce the material of the sticker and the mechanism of absorbing the laser lidar point cloud.

**Details Of Ethics Concerns:**

No.

---

### Official Review · Reviewer_upZA · 2024-11-02

**Soundness:** 2
**Presentation:** 3
**Contribution:** 2
**Rating:** 3
**Confidence:** 5

**Summary:**

The paper presents an adversarial attack scheme called I-ADV, which is the first work to attack 3D object detection models on the intensity channel of point clouds. I-ADV is a gradient-based attack method that updates perturbations based on the gradient sign during iterations. Its main features can be summarized in three points: voxel-level updates, a gradient enhancement technique that combines 3D angle and distance features, and an extremum-based gradient fusion strategy. The voxel-level attack enhances the method's applicability in the real world, while the proposed gradient enhancement technique and gradient fusion strategy improve the performance of I-ADV. In the authors' experimental setup, the latter contributed to an over 40% increase in attack success rate.

**Strengths:**

First, the paper presents the I-ADV adversarial attack scheme, which is the first work to apply adversarial attacks to point cloud intensity, showcasing innovation. Second, the paper makes efforts regarding the feasibility of the attack, including the theoretical possibility demonstrated in Figure 1, the algorithm design for perturbations at the voxel level, and the evaluation of transferability in experiments, all of which enhance the usability of the proposed method. Finally, the writing of the paper is clear and accessible, with excellent explanations from the motivation to the introduction of the method, and the provided conceptual diagrams are easy to understand.

**Weaknesses:**

The main weaknesses of this paper lie in the experimental section. The experiments conducted in the paper are all carried out in the digital domain, and the attack method of "invisible stickers" mentioned in the title is not reflected in the experiments. Moreover, the paper fails to adequately consider the issues encountered in real-world scenarios. In fact, aside from the material issues and multi-sensor problems mentioned by the authors in the limitations section, there are many unavoidable problems, such as the dynamic changes in external lighting and laser incidence angles affecting echo intensity in dynamic environments. Furthermore, I believe there are unreasonable aspects in the experimental setup used in the paper, which include the following concerns:
1.	The use of global perturbations in the experiments raises my concerns about practicality. Although voxel-level perturbations are more realistic than point-level perturbations, controlling the reflection intensity across an entire vehicle is not feasible.
2.	In the main experiments of the paper, the maximum perturbation limit is set to 1, allowing the intensity of all points to vary freely between [0, 1], which further increases my concerns about practicality.
Overall, I believe that what the authors demonstrate is more of a theoretical possibility rather than physical realizability, and I am concerned about the rationality of the experimental setup.

**Questions:**

Apart from about the usability of the methods, my main concerns are as follows:
	The ablation results in Table 3 indicate that extremum-based gradient fusion strategy contributes the most to the results. However, the paper neither explains the motivation for adopting this fusion strategy nor analyzes why it outperforms other strategies. It would be better if the authors could include a discussion on this aspect.
	I noticed that in the visualization of Figure 4, various attack methods tend to extreme the intensity of points, with the sample point cloud primarily consisting of low-intensity points (blue) and high-intensity points (red). Additionally, the greater the number of high-intensity points, the better the performance of the attack. I also found a recent study [1] indicating that increasing the intensity of the point cloud can significantly reduce the detector's confidence. Considering that the experiment allows the intensity of all points to vary freely between [0, 1], I wonder if the significant improvement of I-ADV comes from having more high-intensity points? If so, why not set the intensity of the point cloud directly to 1?
	I observed that Figure 3(a) presents results for smaller voxel sizes, and I estimate from the figure that the minimum setting is approximately 0.015 meters. The issue is that, at such a small voxel size, the vast majority of voxels containing points have only one point. Therefore, in Equation (8), we have g_(V_(m,+) )=g_(V_(m,-) ), leading to g_(V_(m,f) )=+, which indicates that the intensity of most points in the sample will continually increase until it is pushed to 1. This seems to support my observation in point 2. I hope the authors can provide specific settings for the voxel size in Figure 3(a) as well as the visual results under the minimum voxel size setting.
In addition to these, there are a few minor issues:
	Line 248: The statement "We utilize the framework of MI-FGSM to access the gradients during iterative updates" is confusing. I understand it corresponds to line 6 of Algorithm 1, but I do not grasp the meaning of "access" in this context.
	Algorithm 1, line 6: The function L is not explained. I know it represents a loss related to classification, but I do not know what it specifically refers to.
	Figure 3: The x-axis label of the bar chart repeats the subtitles of the subplots.

[1] Sako O, Sato T, Hayakawa Y, et al. Poster: Intensity-Aware Chosen Pattern Injection LiDAR Spoofing Attack[J].

---

### Official Review · Reviewer_bb3M · 2024-11-05

**Soundness:** 2
**Presentation:** 3
**Contribution:** 2
**Rating:** 3
**Confidence:** 2

**Summary:**

This paper proposes an intensity-based voxel-wise attack method called I-ADV for learning-based 3D detectors. I-ADV considers two intensity-related features: 3D angle and distance. The intensity perturbation is optimized using an extremum-based fusion method. The approach is evaluated on the KITTI dataset for the 3D detection of car category and demonstrates superior performance compared to other geometry-based baselines in terms of attack success rate.

**Strengths:**

- The idea of attacking point cloud detectors using intensity information is innovative and thought-provoking.
- The experiments analysis is comprehensive, including an ablation study and evaluations of transferability.
- Experimental results indicate a significantly higher attack success rate compared to the selected baseline methods.

**Weaknesses:**

- A major concern is that the proposed method is only applicable to models that incorporate intensity. It is highly possible that intensity-aware detectors are more vulnerable. Moreover, geometry(xyz)-based models are more popular in practical applications.
- The proposed method has only been evaluated on a single dataset, limiting the generalizability of the findings.
- The method is a white-box attack approach, requiring gradient access to the model, which limits its applicability in practical scenarios.
-  Although three main components are proposed, the contributions of the 3D angle and distance features appear to be minimal compared to the extremum-based fusion approach.
- It lacks evaluations on the more recent transformer-based approach.

**Questions:**

It is unclear how the victim models were trained. Were they simply the official pre-trained models? Additionally, was the intensity feature utilized during the training process? Providing a table detailing the experimental settings would help to clarify these aspects and make the setup easier to understand.

---

### Note · Authors · 2024-11-21

I have read and agree with the venue's withdrawal policy on behalf of myself and my co-authors.